# Study of Mass Transfer Enhancement of Electrolyte Flow Field by Rotating Cathode in Through-Mask Electrochemical Micromachining

**DOI:** 10.3390/mi14071398

**Published:** 2023-07-09

**Authors:** Guoqian Wang, Shan Jiang, Shoudong Ni, Yan Zhang

**Affiliations:** School of Mechanical and Power Engineering, Nanjing Tech University, Nanjing 211816, China; js2282409655@163.com (S.J.); nsd@njtech.edu.cn (S.N.); zhangyanzy@njtech.edu.cn (Y.Z.)

**Keywords:** rotating cathode, through-mask electrochemical micromachining, flow field analysis, efficient mass transfer

## Abstract

To solve the problem of the nonuniform distribution of temperature and electrolytic products in the electrolyte flow field during through-mask electrochemical micromachining, the use of a rotating cathode with surface structures is proposed. The rotation of the cathode increases the efficiency of heat and mass transfer by the electrolyte flow. Simulations are performed to analyze the influence of the type of surface structure, the number of surface structures, and the rotational speed of the cathode on the electrolyte flow field. The results show that the use of a rotating cathode with surface structures significantly improves the mass transfer efficiency of the electrolyte flow field in comparison with a conventional cathode structure, and, in particular, a grooved rotating cathode can increase the outlet flow velocity by about 23%. An experimental demonstration of micropit array processing shows that the use of a grooved rotating cathode increases the mass transfer efficiency by 34% and the processing efficiency by nearly 40% compared with a smooth-surfaced rotating cathode. The grooved rotating cathode also gives the highest machining accuracy. Using this cathode, a uniform micropit array with an average micropit diameter of 201.83 μm, a diameter standard deviation of 3.49 μm, and a depth standard deviation of 0.87 μm is processed.

## 1. Introduction

Surface micro- and nanostructure has an important influence on the surface friction and wettability of mechanical components. There is growing interest in the application of surface structure, especially on the micro- and nanoscale, to advanced techniques in many fields, such as aviation, electronics, energy, optics, machinery, tribology, biology, and bionics [1,2,3]. As a noncontact and nonthermal processing method, electrochemical micromachining is widely used in the fabrication of micro- and nanostructures because of its stress-free and heat-free nature [4,5,6]. The GE and Lehr Precision companies have jointly investigated the multi-axis CNC electrochemical machining of blisks and have successfully developed a five-axis CNC electrochemical machine tool, which greatly shortens processing times and avoids residual stress from machining. The MTU Company has adopted high-frequency narrow pulse vibration electrochemical machining. The Philips Aerospace Company has also used electrochemical machining methods. It has been applied to the blade disk processing of respective military aircraft engines and achieved good results [7]. Natsu et al. [8] used an electrojet technique to limit the electrolytic current to a limited area and successfully processed an array of micropits, each of diameter 300 μm and depth 20 μm. Park et al. [9] used a wire with a diameter of 10 μm as an electrode to electrochemically process a gold sheet to produce a surface structure with submicrometer dimensions. However, there are also some problems with electrochemical micromachining. For example, both single-electrode and electrojet electrochemical machining are point-by-point processing techniques, and their processing efficiency is low.

Through-mask electrochemical machining is based on electrochemical machining and uses lithographic techniques to dissolve a large number of required areas in parallel on a photoresist coated onto the surface of the workpiece. This is an accurate and relatively fast process, and it can produce well-defined surface structures with controlled size, position, and density [10,11,12]. Shenoy and Datta [13] investigated the influence of mask sidewall angle on the anode forming process in through-mask electrochemical machining and found that the use of an angle of less than 90° can enhance the shielding effect of the mask and improve the etching rate. Cai and Li [14] proposed a dry-film electrochemical machining technique using a solid-state photoresist as a mask, which decreased the need for pre-baking, post-baking, leveling, and other operations, thereby reducing labor costs. They were able to process pits with an average diameter of 110 μm and depth 15 μm. However, there are also some problems that need to be solved in mask electrochemical machining. The flow direction of the electrolyte in this technique is perpendicular to the patterned photoresist, and vortices will form in the electrolyte flow. The flow velocity of the electrolyte in the mask hole is very low, which hampers the removal of electrolytic products and of Joule heat. In addition, the nonuniformity of the flow field as the electrolyte passes through the mask leads to poor consistency of the material removal rate and hence poor machining accuracy. Thus, the electrolyte flow field is an important factor affecting the electrochemical machining process. Wang et al. [15] studied methods of improving the flow field during through-mask electrochemical machining and found that the use of a mask with conical holes improved the electrolyte flow velocity. Liu et al. [16] designed a machining device for blade electrochemical machining in which simultaneous side flow and W-type flow generated a three-dimensional composite flow. Their numerical and experimental results showed that this flow mode improved the stability of the flow field in the electrochemical machining area, increasing the mass transfer rate, and improving machining accuracy and quality. They obtained a roughness Ra = 0.4.

In addition to changing the flow field mode, innovative simulation methods also contribute to the optimization of the flow field in electrochemical machining. Dabrowski and Paczkowski [17] carried out a two-dimensional simulation of the electrolyte flow in the machining gap, and compared and analyzed the electrolyte flows in the cases of continuous feed and vibrating feed. Kozak et al. [18] developed engineering software for the simulation of electrolyte flow in different electrochemical machining methods. Fujisawa et al. [19] presented a multiphysics field model for simulating electrochemical machining processes. Tang and Gan [20] used a 3D model to simulate and design a cascade-channel processing cathode. The electrolyte flow field directly affects the stability and processing quality of through-mask electrochemical machining. An unsuitable electrolyte flow field will lead to poor processing quality and short-circuiting.

In the above literature, the researchers optimized the flow field in electrochemical machining with regard to different aspects, such as the mask, processing device, and processing electrode, so as to ensure the high-speed flow of electrolyte in the interelectrode gap during the electrochemical machining process, take away the electrolytic products (gases, quantity of heat), enhance the conductivity and distribution more evenly, and accelerate the electrochemical reaction rate. This paper focuses on the influence of the different types of cathode structures, especially the rotational speed and surface structure (grooved, toothed, and blade-like) of a cathode consisting of rotating cylinders on the electrolyte flow field, with the aim of obtaining the optimum flow field for through-mask electrochemical machining.

## 2. Basic Principle of Rotating Cathode Through-Mask Electrochemical Micromachining

The electrolyte flow in the gap in the electrochemical machining process not only promotes the electrochemical reaction at the anode workpiece, but also takes away the electrolytic products, thereby ensuring the smooth progress of electrochemical machining and the accuracy of the machined workpiece. Gases and heat generated by electrolysis will affect the conductivity of the electrolyte, and the conductivity determines the rate of the electrochemical reaction. During electrolysis, an uneven distribution of gas void fraction and temperature will lead to a nonuniform conductivity distribution, which will increase the machining error. The distribution of gas and temperature in the electrolyte flow field is calculated and analyzed through numerical simulation under different cathode structures. The assumptions during the simulation process are as follows: (1) Since the main gas product is insoluble hydrogen gas, the electrolyte flow is considered as a gas–liquid two-phase flow; (2) Due to the small resistance of the metal electrode relative to the electrolyte, the Joule heat generated by the metal electrode is ignored; (3) The heat in the flow of electrolyte is mainly convective heat transfer, so the radiative transfer and thermal conductivity of heat are ignored. The numerical analysis was conducted using COMSOL software, and the main mathematical models are as follows.

In the process of electrochemical machining, metal cations and insoluble electrolytic products will be generated on the anode surface, and hydrogen will be generated on the cathode. Due to the fact that the volume of gas products is much larger than that of other insoluble products, the flow of electrolyte can be regarded as gas–liquid two-phase flow. The gases in the electrolyte are considered to have no slip between them, so the flow of the electrolyte can be represented by the Navier Stokes equation:(1)$$\rho \frac{\partial u}{\partial t}+\rho \left(u\cdot \nabla \right)u=\nabla \left[-\bar{p}+\mu \nabla u\right]+F_{t}$$
where *u* is the electrolyte flow velocity, ρ indicates the density of the electrolyte containing gas, *μ* is the viscosity coefficient, $\bar{p}$ is the average pressure in the flowing electrolyte, and $F_{t}$ is external force.

In the electrolytic reaction simulation, the electric field in the electrolyte is considered as a passive steady current field, and its potential distribution meets the Laplace’s equation, that is
(2)$$i_{e}=\kappa \cdot \nabla \Phi$$
where $i_{e}$ is the current density within the electric field area, Φ is the potential in the electric field region, and *κ* is the conductivity of the electrolyte.

The hydrogen gas released from the cathode during the electrolysis process and the heat generated by electrolysis will affect the conductivity of the electrolyte:(3)$$\kappa =\kappa_{0}\cdot {\left(1-\beta \right)}^{n}\left[1+\gamma \left(T-T_{0}\right)\right]$$
where $\kappa_{0}$ is the initial conductivity of the electrolyte, *β* is the volume fraction of the gas phase in the electrolyte, *n* is the Brugmann coefficient, usually taken as 1.5, $T_{0}$ is the initial temperature of the electrolyte, *T* is the actual temperature of the electrolyte, and *γ* is the temperature influence coefficient.

The heat generated during electrochemical machining is mainly composed of the Joule heat of the electrolyte and the electrochemical reaction heat at the edge of the electrolyte electrode.
(4)$$\mathrm{Joule}\;\mathrm{heat}\;Q_{j}=\kappa {\left(\nabla \varphi \right)}^{2}$$
(5)$$\mathrm{Electrochemical}\;\mathrm{reaction}\;\mathrm{heat}\;Q_{r}=U_{k}\cdot i_{k}$$

The electrolyte temperature distribution can be described by the fluid convection diffusion equation:(6)$$\rho \cdot C_{p}\left(\frac{\partial T}{\partial t}+\left(u\cdot \nabla \right)T\right)=\nabla \left(\kappa \nabla T\right)+Q_{K}$$
(7)$$Q_{K}=Q_{j}+Q_{r}$$
where $C_{p}$ is the specific heat capacity of electrolyte at normal pressure, *κ* is the conductivity of the electrolyte, *T* is the temperature of the electrolyte, $Q_{K}$ is the heat generated by electrochemical machining, $Q_{j}$ is joule heat, and $Q_{r}$ is the heat of electrochemical reaction.

In through-mask electrochemical micromachining, all cathode currents are used to generate hydrogen gas, and the current density efficiency is set to 100%. The hydrogen flux generated by the cathode reaction meets the following equation:(8)$$R_{i}=\frac{i_{c}}{z_{i}F}$$
where $R_{i}$ is the hydrogen flux, $i_{c}$ is the cathode current density, $z_{i}$ is the number of ion charges, and *F* is the Faraday‘s constant.

Figure 1a is a schematic of a flat cathode electrochemical machining process in which the electrolyte flow in the interelectrode gap is side flow. Based on the various parameters of the experimental platform, the simulation conditions are as follows: the initial temperature is 20 °C, the processing voltage is 20 V, the interelectrode gap is 1 mm, the inlet flow velocity is 0.1 m/s, and the initial gas void fraction is 0. The distributions of the gas void fraction and temperature in the processing zone are shown in Figure 1b,c, respectively. It is clear that the gas aggregate and the temperature rises along the direction of electrolyte flow. According to the curves, the gas void fraction increased to 12% and the temperature increased by 7 K. The accumulation of gas and temperature can affect the conductivity of the electrolyte, which in turn has very adverse effects on the consistency of machining accuracy.

For this reason, a reverse flow is adopted for the electrolyte, and the structure of the cathode is changed. The basic principle of this approach is shown in Figure 2. Here, the flat cathode is replaced by a cathode consisting of a large number of rotating cylinders separated by small gaps, so that the electrolyte flow is dispersed to a large number of distributed outlets, enabling the electrolytic products to be removed in a uniform manner and reducing the nonuniformity of their distribution on the surface of the workpiece. At the same time, the rotation of the cylinders comprising the cathode enhances the mass and heat transfer efficiency of the electrolyte fluid, thereby improving the processing quality.

## 3. Flow Field Simulation and Analysis of Results

### 3.1. Flow Field Analysis of Smooth-Surfaced Rotating Cathode

In order to analyze the influence of cathode structure on the distribution of electrolytic products, the simulation conditions such as interelectrode gap, inlet flow velocity, and initial temperature are all the same as those in Figure 1. Figure 3 shows the results of flow field simulation using the mathematical model mentioned earlier for the case of a rotating cathode consisting of a large number of rotating smooth-surfaced cylinders. The electrolyte enters from both sides of the inlet and flows out in the middle. The large number of gaps between the cylindrical elements of the rotating cathode and the corresponding microslits above them provide a high flow velocity of the electrolyte and facilitate the swift removal of gases generated in the electrolytic process from the electrolytic reaction zone near the anode, as can be seen in Figure 3a. In addition, it can be seen from Figure 3b that the uniformity of the temperature distribution in the processing area is greatly improved in comparison with that in the case of a static flat cathode, although there is still a large local temperature rise near the middle of the cathode.

The use of a rotating cathode can effectively reduce the accumulation of electrolytic products in the processing zone, but there are still problems of low gas transfer efficiency and high local temperature. To solve these problems, we investigate the effects of the surface structure of the cylinders comprising the rotating cathode and their rotational speed on the flow field of the electrolyte. If this flow field provides a sufficiently high outlet flow velocity of electrolyte, this will facilitate the rapid removal of gas and other electrolytic products from the processing region and the rapid replenishment of the electrolyte in this region, as well as reducing the local temperature rise caused by electrolysis. In this paper, the flow field is simulated for three surface structures on the cylinders of the rotating cathode, namely, grooved, toothed, and blade-like shapes. By comparing results for different rotational speeds of the cathode and for the different surface structures, we select the most suitable combination of surface structure and rotational speed.

In the experiment, we separate multiple pairs of rotating cylinders with baffles, so that each pair of rotating cylinders and baffles form a single independent cavity, and to facilitate the simulation, we simplify the flow field model and consider just one such cavity. Figure 4 shows a simplified flow field models for the different cylinder surface structures. In each case, the diameter of the cylinders is 10 mm, and the right and left cylinders rotate clockwise and counterclockwise, respectively. To ensure comparability with the previous simulation, the inlet velocity at both ends is 0.1 m/s, the outlet pressure is 0, and a turbulent physical field is assumed. The other simulation conditions are as follows: the interelectrode gap is 1 mm, the initial temperature is 20 °C, the processing voltage is 20 V, the inlet flow velocity is 0.1 m/s, and initial gas void fraction is 0.

### 3.2. Influence of Surface Structure of Rotating Cathode on Flow Field

Figure 5 shows the simulated flow velocity distributions of the rotating cathodes with grooved, toothed, and blade-shaped surface structures both when the cathode is stationary and when it is rotating. The arrows in the Figure 5 indicate the flow direction of the electrolyte. It can be seen that compared with the stationary case, when the cathode rotates, the flow velocity in the region between the two cylinders, i.e., in the electrolytic processing area, is significantly increased, from about 0.7 m/s to about 2.0 m/s, which is more than double. This region is directly connected to the outlet, and so the outlet flow velocity is also increased. Compared with flat cathode, the flow direction of the electrolyte has undergone significant changes, and deviation from the processing zone is more conducive to product transport. More importantly, the outlet flow velocity has increased by 186%. Thus, the use of a rotating cathode with surface structure provides a stable flow field in the electrolytic processing area, with a high flow velocity and a high mass transfer efficiency.

Figure 6 shows the simulated gas void fraction and temperature distributions for rotating cathodes with the three different surface structures at a rotational speed of 40 rev/s. In the traditional flat cathode ECM, gas and temperature gradually increase along the anode length and accumulate at the outlet, thus affecting the electrochemical reaction. The presence of a structure on the cathode surface significantly accelerates the transfer of gas and heat when the cathode is in the rotating state. This leads to more uniform distributions of the gas void fraction and temperature than in the case of a smooth-surfaced rotating cathode and thus to better electrochemical machining results. For all three surface structures of the rotating cathode, both the flow velocity and electrolytic product distributions are clearly improved in comparison with the case of a smooth surface. The trend of the flow velocity distribution is consistent with that of the electrolytic product distribution. In other words, the increase in flow velocity leads directly to the improved efficiency of mass transfer of electrolytic products. Therefore, a quantitative analysis of the electrolyte flow velocity will be carried out with the aim of selecting the most suitable surface structure and rotational speed for the cathode.

### 3.3. Effect of Rotational Speed on Electrolyte Flow Velocity

To facilitate the analysis of the influence of the rotational speed of the cathode on electrolyte flow, the four positions shown in Figure 7 are selected as observation points of the electrolyte flow velocity. Two points A and B reflect the flow field changes in the processing area. Point C is used to analyze the direct effect of rotational speed on the flow field. Point D gives the flow velocity at the outlet, which is used to analyze the changes in mass transfer efficiency.

The changes in flow velocity changes at points A and B are shown in Figure 8a,b. Different types of surface structure have different effects on the flow velocity, with the most obvious enhancement of flow velocity being obtained with the grooved structure. No matter which structure is used, the rotational speed of the cathode has little effect on the flow velocity at points A and B. The plot of the flow velocity at point C in Figure 8c shows that the electrolyte flow velocity increases proportionally to the rotational speed, with the rate of increase being greater for the grooved structure than for the toothed and blade-shaped structures. The maximum flow velocity for the grooved structure is 3.14 m/s, which is an increase of 288.60% compared with the flow velocity obtained when the rotating cathode is stationary. The dependence of the flow velocity at point D on rotational speed, shown in Figure 8d, is different. In the range of rotational speeds from 0 to 40 rev/s, the flow velocity decreases with increasing rotational speed, whereas in the range from 40 to 80 rev/s, it increases rapidly. For the grooved rotating cathode, the electrolyte outlet flow velocity reaches 1.0 m/s, which is an increase by 23.03% compared with the flow velocity obtained when the rotating cathode is stationary.

It can be seen from Figure 8 that at all the observation points, the effects of the different structure types on the flow field are consistent. That is, the enhancement of the flow velocity by the grooved structure is the strongest, the enhancement by the blade-shaped structure is the weakest, and the enhancement by the toothed structure is intermediate. On the other hand, the effect of the rotational speed of the cathode differs at the different observation points. The rotational speed has little effect on the flow velocity at points A and B, but has an obvious effect at points C and D. The variations in flow velocity at the different observation points indicate that the surface structure of the rotating cathode has a global effect on the flow field, whereas the effect of the rotational speed on the flow velocity is localized to the regions that are close to the outlet.

There are also differences in the effect of the rotational speed at points C and D. These are due to different effects of the rotating cathode on the electrolyte flow at these points. Point C is located between the two rotating cylinders, the rotation of which acts directly on the electrolyte flow, and so the flow velocity increases proportionally to the rotational speed. However, at point D, which is above the cathode, the effect of cathode rotation depends on the flow velocity. In the range of rotational speeds between 20 and 40 rev/s, the linear velocity of the cathode surface is 0.31–0.63 m/s, which is lower than the electrolyte flow velocity observed at points A and B. Since the linear velocity of the cathode surface is lower than the electrolyte flow velocity, it is not enough to affect the direction of fluid flow at long distances. On the contrary, under the action of centrifugal force, the fluid near the cathode surface is deflected to both sides of the fluid outlet, and so the flow velocity at the outlet is lower than that when the cathode is stationary. At rotational speeds of 40–80 rev/s, the linear velocity of the surface is 0.63–1.26 m/s, which exceeds the original velocity of the electrolyte flow and affects the flow at long distances. Therefore, the electrolyte will flow to the outlet with a higher flow velocity at high rotational speeds.

The above analysis of the flow velocity and the efficiency of mass transfer of electrolytic products indicates that a high flow velocity improves the removal of gas and heat, enhances the efficiency of mass transfer, and improves the accuracy and efficiency of electrochemical machining.

### 3.4. Effect of Number of Surface Structures on Outlet Flow Velocity of Electrolyte

The number of structures on the surface of the rotating cathode may be another key parameter affecting mass transfer. Therefore, in this subsection, the flow velocity is investigated for different numbers of surface structures: 8, 16, and 24. A rotational speed of 80 rev/s is chosen, since the preceding simulation results have shown that this gives a high flow velocity and enhanced mass transfer efficiency of electrolytic products. Figure 9 shows the effect of the number of surface structures on the flow velocity of electrolyte at the outlet point D. It can be seen that the flow velocity increases with an increasing number of structures. The greater the number of surface structures, the more effectively can the flow of electrolyte be driven by the rotating cathode, thus giving a higher outlet flow velocity. However, the number of surface structures cannot be increased indefinitely. As the number of structures continues to increase, if the diameter of the rotating cylinder is not to be increased (which of course is not desirable), then the size of each structure must be reduced, which is not conducive to improving the flow field. For 8–16 surface structures, the electrolyte outlet flow velocity rises very slowly, but when there are 16–24 structures, it rises much more rapidly. The outlet flow velocity of a rotating cathode with 24 surface structures is increased by 26.02% compared with that of a cathode with eight structures. The flow field is more stable and provides more efficient mass transfer.

Considering the influence of the type and number of surface structures and the rotational speed on the outlet flow velocity of electrolyte at point D, it was found that a grooved rotating cathode provides the greatest enhancement of the flow field and that the higher the rotational speed, the more efficient is the mass transfer. The outlet flow velocity of the electrolyte increases with an increasing number of surface structures, and, if the diameter of the rotating cathode is not to be changed, the maximum number is found to be 24. Thus, in summary, the simulation results show that a grooved cathode with 24 surface structures and a rotational speed of 80 rev/s gives the optimum flow velocity and efficiency of mass transfer.

## 4. Experiments

To verify the feasibility of a rotating cathode for practical application in through-mask electrochemical machining, an experimental study of the fabrication of a metal micro-pit array was carried out. The standard of micro-pit machining experiment is that the diameter of pit is 200 ± 10 μm, and the depth is 10 μm to 15 μm. The cathode cylinders and the experimental device are shown in Figure 10. The device included a clamp, motor, transmission device, pulsed power supply, and electrolyte circulation system. Multiple pairs of rotating cathodes were placed in the closed cavity formed by the clamping body and the workpiece, and separated by baffles. The output shaft of the motor was connected with the input shaft of the transmission device, the output shafts of which were connected with the rotating cathodes and drove their rotation. In the electrochemical machining process, the electrolyte entered the electrolytic processing area from the liquid supply pipeline, the electrolyte and electrolytic products entered the return tank from the return pipeline, and the electrolytic products were filtered and removed through the filtration device to ensure that the electrolyte in the liquid supply tank remained clean. The simulation results showed that the optimum heat and mass transfer efficiencies were obtained for 24 surface structures on a cathode with a rotational speed of 80 rev/s. To simplify the experimental procedure, these values for the number of surface structures and the rotational speed were adopted throughout the experiments. Processing experiments were carried out for all three kinds of surface structure, with the experimental parameters shown in Table 1.

The analysis of processing quality was mainly carried out from two aspects: measurement of microstructure size and observation of surface morphology. Sixteen detection points were randomly selected on the processed workpiece anode. The diameter and depth of the micropits were measured using a laser confocal microscope, and the average value and standard deviation value were calculated using the formulas shown in Table 2. The standard deviation of micropit size characterizes the uniformity in size of the micropits in the array and the machining accuracy of the array. The smaller the standard deviation, the better the uniformity of size and the higher the machining accuracy.

In addition to these measurements, the proportion of solid products in the solution discharged from the outlet was measured. The processing time was taken to be the time at which the micropit depth reached 1 μm.

Figure 11a shows the statistical results for processing quality. Compared with the unstructured rotating cathode, the flat-cathode gave smaller standard deviations of micropit diameter and depth. Although the unstructured rotating cathode improves the flow field, as evidenced by the higher content of solid products in Figure 11b compared to the flat-cathode, the curved cathode interface also exacerbates the differences in forming dimensions. However, when the surface of the rotating cathode has a structure that further enhances the mass transfer of the convective field, smaller standard deviations of diameter and depth are obtained. Thus, the use of a surface structure on the rotating cathode can effectively improve processing quality, with the grooved rotating cathode giving the best results. The statistical results in Figure 11b show that the mass transfer efficiency and processing efficiency of the rotating cathode are higher than those of the flat cathode. Meanwhile, the processing efficiency and mass transfer efficiency depend on the type of surface structure. The cathodes with surface structures require a shorter processing time to obtain a given micropit size, and the proportion of solid products in the discharged solution is greater with these cathodes, indicating that surface structures on a rotating cathode increase both production efficiency and the efficiency of mass transfer of electrolytic products, leading to improved processing quality. It is found that of the three types of surface structure, the grooved structure gives the best results. Thus, the experimental results are consistent with those from the simulation.

To further quantify the results of processing using the grooved rotating cathode, Table 3 lists the micropit diameter and depth values at 16 sample points on the processed micropit array. The are only small variations in the diameters and depths of the micropits in the array, with the diameters all lying in the range 196.17–206.49 μm and the depths in the range 11.45–14.65 μm. The average diameter is 201.46 μm, with a standard deviation of 3.49 μm, and the average depth is 13.15 μm, with a standard deviation of 0.87 μm. The standard deviation of the diameter is rather large, indicating that the machining of the micropit diameter is less consistent than that of the depth. There are a number of reasons for this. On the one hand, in the electrochemical machining process, the edge effect of the electric field makes the current density of the anode at the edge of the mask hole higher than that in other areas, resulting in increased lateral corrosion of the micropit in the radial direction. This increase in current density is not uniform, leading to fluctuations in the micropit diameter. In addition, some of the micropits have diameters smaller than the supposed 200 μm diameter of the mask through-holes. This is because uneven exposure and development during the mask preparation process led to the production of a dry film colloid residue, with the result that some mask holes were actually less than 200 μm in diameter. More accurate measurements of the diameter of the micropits in the array showed that the maximum diameter of the micropits was 207.33 μm and the minimum diameter 195.80 μm, and although the standard deviation of the micropit diameter was still rather large, the maximum deviation was 11.53 μm, which represents high machining accuracy.

Figure 12 shows a scanning electron microscope (SEM) image, three-dimensional contour map, and cross-section of the micropit array obtained using the grooved rotating cathode. From the SEM image in Figure 12a, it can be seen that the rounded morphology of the array is good, the mask confinement is excellent, and the non-processed area is well protected. In addition, it can be seen from the contour map in Figure 12b and the cross-section in Figure 12c that the side walls of the micropits are steep and their bottom nearly flat. This shows that the morphology of the fabricated micropit array is good. On the basis of the above measurements and observations, it can be concluded that the grooved rotating cathode can process an array of flat-bottomed micropits with good consistency.

## 5. Conclusions

To solve the problem of gas agglomeration and high local temperatures and optimize the flow field in through-mask electrochemical machining, this paper has proposed a machining method using a rotating cathode with surface structure. The effects of the rotational speed of the cathode and different types of surface structure on the electrolyte flow field have been simulated and analyzed. Finally, a micropit processing experiment has been carried out on the surface of 304 stainless steel using a grooved rotating cathode. The following conclusions can be drawn from the results of the simulation and experiment:In contrast to processing using a side-flow flat cathode, the rotating cathode splits the electrolyte flow into multiple outlets, giving a flow field that is conducive to the removal of bubbles and other electrolytic products.The optimum flow field is obtained using a rotating cathode with 24 surface grooves and rotating at 80 rev/s, for which the flow velocity at the electrolyte outlet is increased by 23.0% compared with a stationary smooth-surfaced cathode. Not only does the outlet flow velocity increase, but also a more stable flow field is obtained and the efficiency of heat and mass transfer is increased.The experimental demonstration of the proposed method produced a surface micropit array with a maximum micropit diameter of 207.33 μm, a minimum diameter of 195.80 μm, an average diameter of 201.46 μm, and a depth standard deviation of 0.87 μm.

## Figures and Tables

**Figure 1 micromachines-14-01398-f001:**
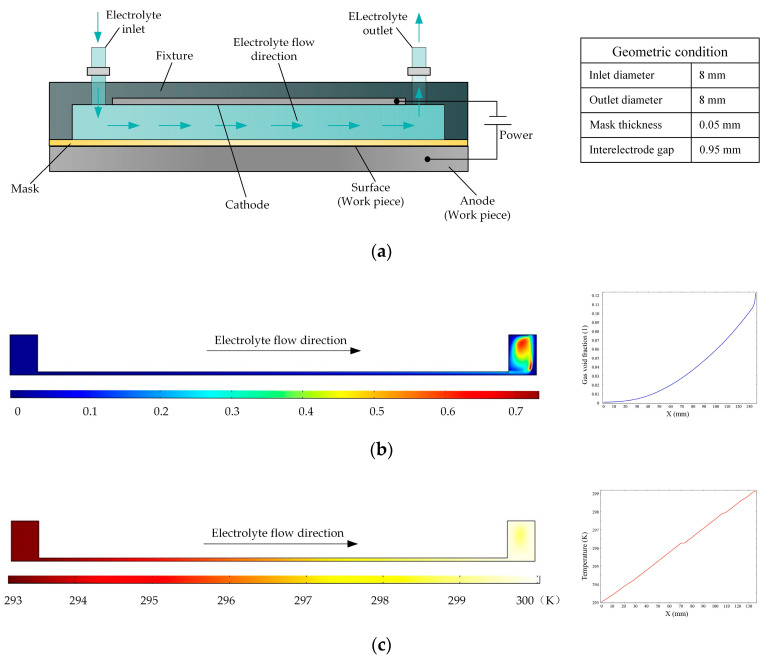
(**a**) Schematic of the electrochemical machining process with a flat cathode; (**b**,**c**) gas void fraction and temperature distributions, respectively, in flat cathode electrochemical machining.

**Figure 2 micromachines-14-01398-f002:**
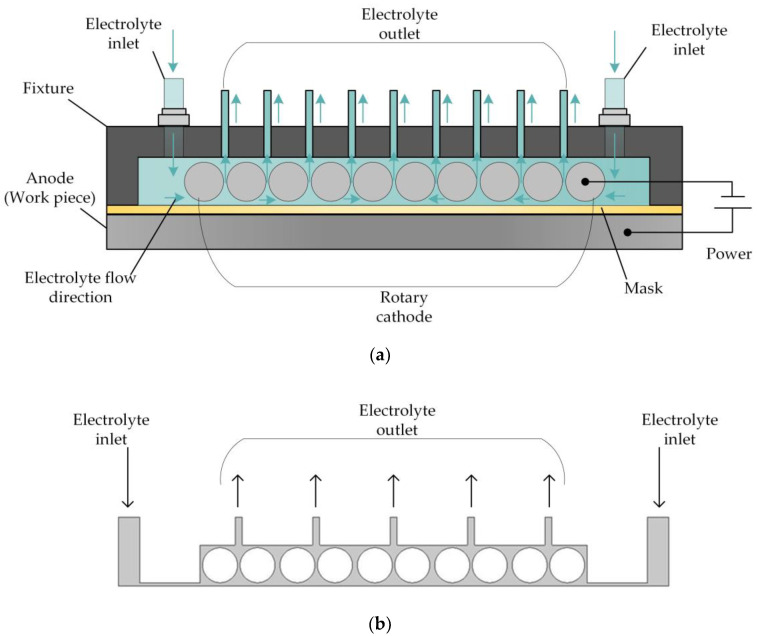
(**a**) Schematic of rotating cathode electrochemical machining. (**b**) Flow field in rotating cathode electrochemical machining.

**Figure 3 micromachines-14-01398-f003:**
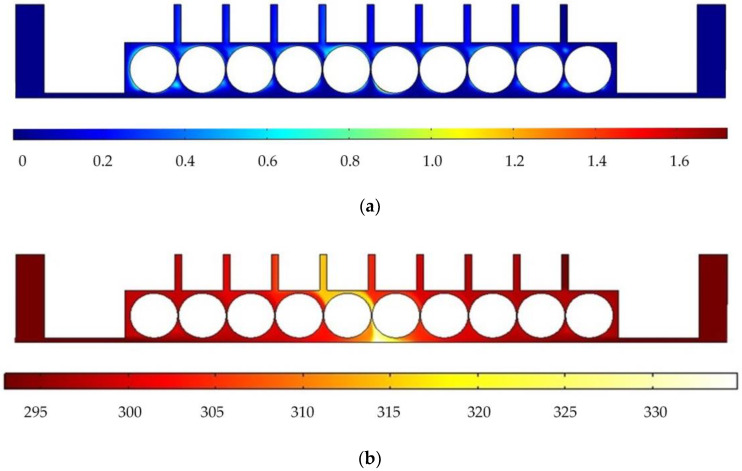
Simulation results for rotating cathode electrochemical machining: (**a**) distribution of gas void fraction; (**b**) temperature distribution.

**Figure 4 micromachines-14-01398-f004:**
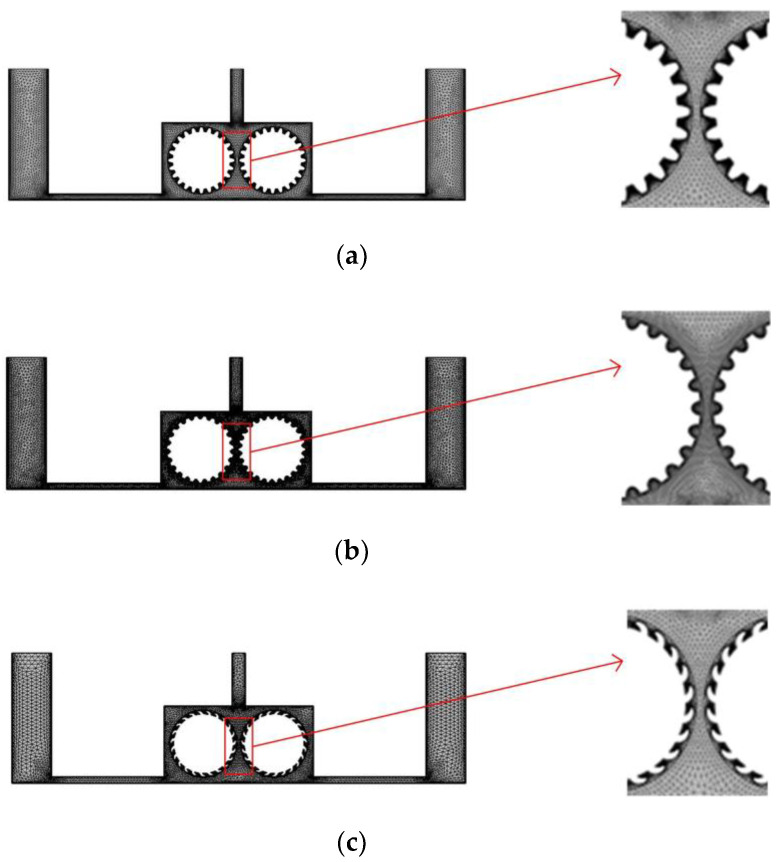
Grid diagrams for the rotating cathodes with different surface structures: (**a**) grooved; (**b**) toothed; (**c**) blade-shaped.

**Figure 5 micromachines-14-01398-f005:**
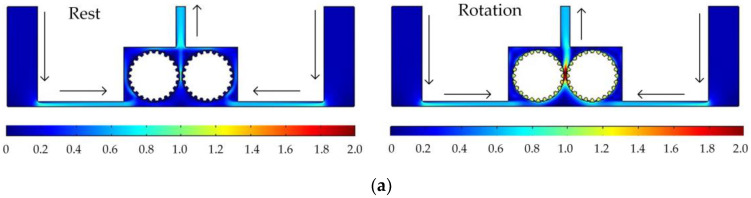
Simulated flow velocity distributions for rotating cathodes, both at rest and in rotation, with different surface structures: (**a**) grooved; (**b**) toothed; (**c**) blade-shaped.

**Figure 6 micromachines-14-01398-f006:**
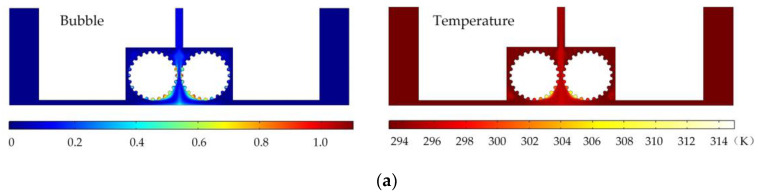
Simulated gas void fraction and temperature distributions for rotating cathodes with different surface structures: (**a**) grooved; (**b**) toothed; (**c**) blade-shaped.

**Figure 7 micromachines-14-01398-f007:**
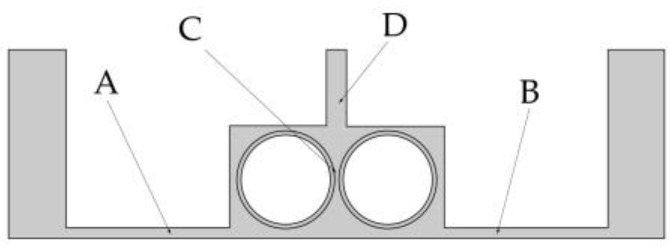
Four observation points in the flow field model.

**Figure 8 micromachines-14-01398-f008:**
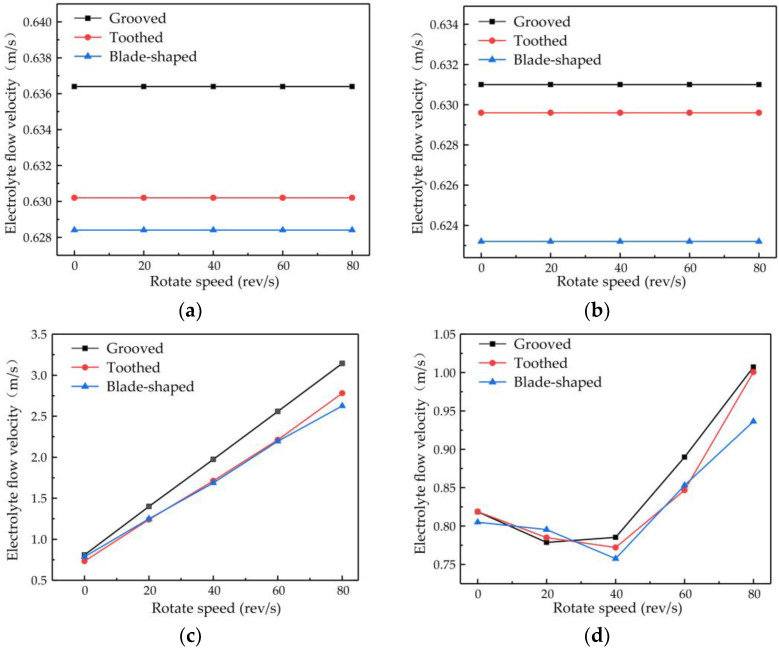
Flow velocity versus rotational speed of rotating cathodes with different surface structures at different observation points: (**a**) point A; (**b**) point B; (**c**) point C; (**d**) point D.

**Figure 9 micromachines-14-01398-f009:**
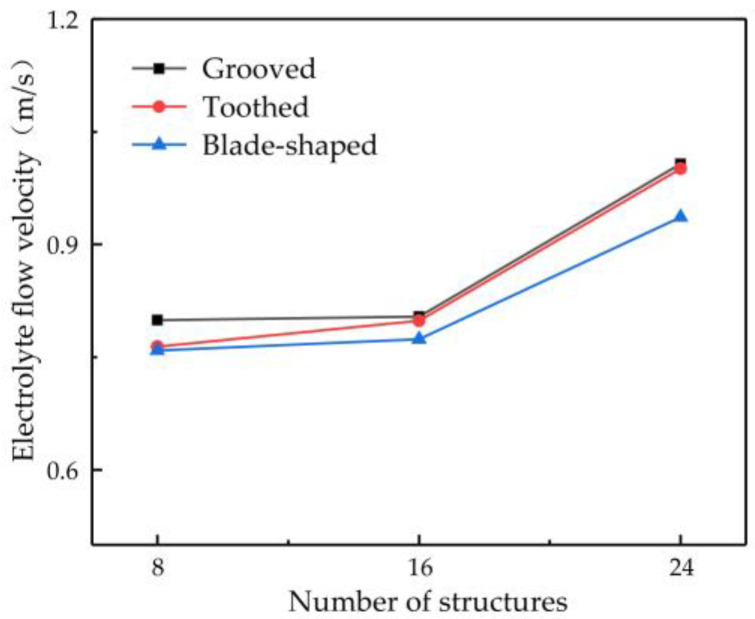
Relationship between number of surface structures and flow velocity at observation point D.

**Figure 10 micromachines-14-01398-f010:**
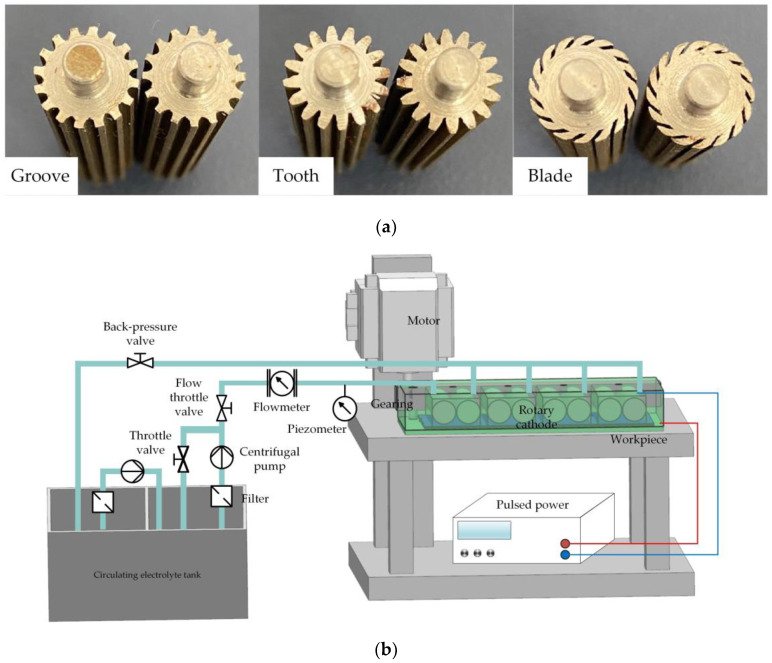
(**a**) Cylinders of rotating cathodes with grooved, toothed, and blade-shaped surface structures. (**b**) Experimental device.

**Figure 11 micromachines-14-01398-f011:**
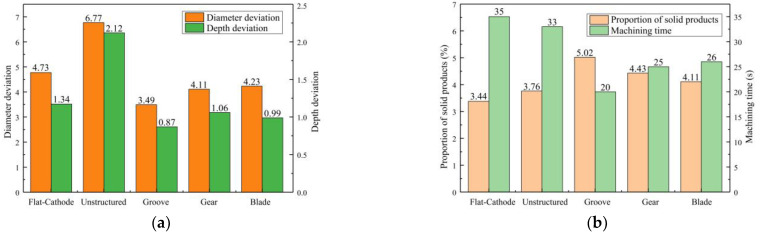
Statistical results for machining quality of different cathode structures: (**a**) Standard deviation of micropit diameter; (**b**) Processing time and proportion of solid products.

**Figure 12 micromachines-14-01398-f012:**
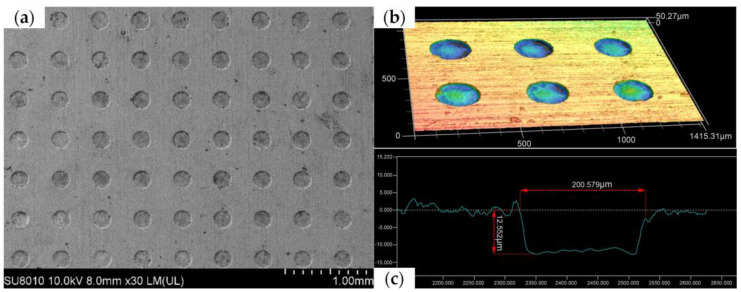
Different surface structures: (**a**) SEM image, (**b**) three-dimensional contour map, and (**c**) cross-section of micropit array processed by grooved rotating cathode.

**Table 1 micromachines-14-01398-t001:** Experimental parameters.

Parameter	Values
Electrolyte concentration *c*	200 g/L NaNO_3_
Electrolyte temperature *T*	25 °C
Mask hole diameter *D*	200 μm
Mask hole length *l*	400 μm
Workpiece anode material	Stainless steel 304
Mask thickness *H*	50 μm
Interelectrode gap *g*	1 mm
Voltage *U*	20 V
Frequency f	4 KHz
Pulse duty factor	20%
Inlet velocity *v*	0.5 m/s
Outlet pressure *P*	0

**Table 2 micromachines-14-01398-t002:** Formulas for average and standard deviation of micropit dimensions.

Parameter	Formula
Average diameter of micropits, *d*	$d=\sum_{i=1}^{N}\frac{d_{i}}{N}$
Average depth of micropits, *h*	$h=\sum_{i=1}^{N}\frac{h_{i}}{N}$
Standard deviation of micropit diameter, *S_d_*	$S_{d}=\sqrt{\sum_{i=1}^{N}\frac{{\left(d_{i}-d\right)}^{2}}{N-1}}$
Standard deviation of micropit depth, *S_h_*	$S_{h}=\sqrt{\sum_{i=1}^{N}\frac{{\left(h_{i}-h\right)}^{2}}{N-1}}$

*N* is the total number of micropits measured, *d_i_* is the diameter of the *i*th micropit measured, and *h_i_* is the depth of the *i*th micropit measured.

**Table 3 micromachines-14-01398-t003:** Measurements of micropits.

Sample Point	1	2	3	4	5	6	7	8
*d* (μm)	205.16	204.45	197.00	206.49	196.17	202.98	206.35	203.36
*h* (μm)	14.65	13.67	13.98	13.18	14.09	13.69	13.34	13.72
Sample point	9	10	11	12	13	14	15	16
*d* (μm)	196.42	199.75	201.62	198.64	203.55	203.01	202.00	196.40
*h* (μm)	13.79	12.87	12.75	11.51	12.59	12.57	11.45	12.49

## Data Availability

Not applicable.

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
