# Peer review of "Study of Mass Transfer Enhancement of Electrolyte Flow Field by Rotating Cathode in Through-Mask Electrochemical Micromachining"

_micromachines, 2023, doi:10.3390/mi14071398_

Round 1

Reviewer 1 Report

The manuscript is devoted to study of the through-mask electrochemical micromachining. The research topic is relevant. However, there are a number of comments, so the manuscript cannot be recommended for publication in the presented form.

When preparing a revised manuscript, the following comments should be taken into account.

1. It is recommended to change Fig.1, since:

(1.1) the value of the interelectrode gap (IEG) in the scheme (Fig. 1a) differs significantly from the value of IEG for temperature and gas void fraction distributions (Fig. 1b, с);

(1.2) the temperature and gas void fraction distributions depend on IEG and working conditions, the values of which are not shown in Fig. 1;

(1.3) the temperature and gas void fraction values in the tanks located at the inlet and outlet (Fig. 1b, с) should not have a noticeable effect on the distribution of current density along the length anode;

(1.4) the legend for temperature should start with 293, not 0;

(1.5) judging by (Fig.1b, с) the main changes in temperature and gas void fraction occur along the length, so it is better to show the curves of temperature and gas void fraction distributions on the anode surface in Fig.1b,d;

(1.6) it is not specified which part of the workpiece surface is being processed.

2. Usually, the term “gas void fraction” is used to quantify the concentration of gas bubbles. In this paper, the terms “bubble”, “bubble distributions” (Fig. 1, 6), “distribution of electrolytic products" (Fig. 3), the meanings of which are not defined.

3. In section 3, it is recommended to give the equations of the transfer processes included in the mathematical model with the corresponding initial and boundary conditions.

4. In section 3, it is recommended to present the simulation results for the traditional flat-cathode ECM scheme and compare these results with similar ones for the proposed scheme.

Reviewer 2 Report

The paper titled " Study of Mass Transfer Enhancement of Electrolyte Flow Field by Rotating Cathode in Through-Mask Electrochemical Micromachining " proposes a new method for through-mask electrochemical micromachining, which is very interesting. The rotating cathode has changed the structure of traditional TMECM and reduced the machining gap. The structure of the cathode surface enhances the mass transfer effect. I think the paper can be accepted for publication, but minor modifications are required before that.

1. In the introduction section, the author cited a large number of references, but it should not only list the content of the literature, but also highlight the relationship between the literature and the paper.

2. The simulation results are shown in Figure 1 on the page 3, but there are no volume simulation conditions in the main text. The same problem also appears in the simulation results in Figure 3.

3. On page 6, lines 156 to 157, it is written "with a high flow velocity and a high mass transfer efficiency." Why? Have you considered the adhesion of electrolytic products on the electrode surface?

4. On page 7, Figure 6 shows the simulation results of bubbles and temperature. Are these results obtained under cathode rotation or in a stationary state? If it is in a rotating state, what is the rotational speed?

5. How was the data on solid products obtained in Figure 11 on page 12?

6. Is the description of the standard deviation and deviation of diameter contradictory on page 13, lines 330 to 332? Additionally, what are the criteria for determining machining accuracy?

The English is  acceptable, and no specific comments.

Round 2

Reviewer 1 Report

In the revised manuscript, comments (1) and (2) are only partially taken into account, and comments (3) and (4) are not taken into account at all. The absence of equations describing the mutual influence of the solution flow and transfer processes (charge, heat, gas, etc.) does not allow us to assess the reliability of the results obtained during modeling. Different models of the system under consideration can be used in Comsol, so the equations must be included in the manuscript. In particular, it is completely unclear how the stationary distributions of temperature and gas phase were obtained when processing in the pulse mode (Table 1). If the simulation was performed at constant voltage, then it is necessary to explain how these results are related to experimental data. There is no justification for the choice of the inlet velocity and the interelectrode gap. With an increase in inlet velocity and interelectrode gap, changes in temperature and void gas fraction decrease, which leads to an increase in processing accuracy with a slight decrease in metal removal rate. The results shown in Fig. 11 should be supplemented with data for ECM with a flat cathode.

When preparing the revised version of the manuscript, the comments given earlier should be taken into account, as well as those indicated in this review.

Round 3

Reviewer 1 Report

The revised manuscript mainly takes into account the comments. The manuscript may be recommended for publication in the present form.